# Glucagon-like Peptide-1 Receptor (GLP-1R) Signaling: Making the Case for a Functionally G_s_ Protein-Selective GPCR

**DOI:** 10.3390/ijms26157239

**Published:** 2025-07-26

**Authors:** Anastasios Lymperopoulos, Victoria L. Altsman, Renee A. Stoicovy

**Affiliations:** Laboratory for the Study of Neurohormonal Control of the Circulation, Department of Pharmaceutical Sciences (Pharmacology), Barry and Judy Silverman College of Pharmacy, Nova Southeastern University, Fort Lauderdale, FL 33328-2018, USA; va566@mynsu.nova.edu (V.L.A.); rs2981@mynsu.nova.edu (R.A.S.)

**Keywords:** agonist potency, βarrestin signaling, biased signaling, cyclic adenosine monophosphate, desensitization, glucose-dependent insulinotropic peptide receptor, glucagon-like peptide-1 receptor, glucagon family receptor poly-agonist, G protein signaling

## Abstract

Spurred by the enormous therapeutic success of glucagon-like peptide-1 receptor (GLP-1R) agonists (GLP1-RAs) against diabetes and obesity, glucagon family receptor pharmacology has garnered a tremendous amount of interest. Glucagon family receptors, e.g., the glucagon receptor itself (GCGR), the GLP-1R, and the glucose-dependent insulinotropic peptide receptor (GIPR), belong to the incretin receptor superfamily, i.e., receptors that increase blood glucose-dependent insulin secretion. All incretin receptors are class B1 G protein-coupled receptors (GPCRs), coupling to the G_s_ type of heterotrimeric G proteins which activates adenylyl cyclase (AC) to produce cyclic adenosine monophosphate (cAMP). Most GPCRs undergo desensitization, i.e., uncouple from G proteins and internalize, thanks to interactions with the βarrestins (arrestin-2 and -3). Since the βarrestins can also mediate their own G protein-independent signaling, any given GPCR can theoretically signal (predominantly) either via G proteins or βarrestins, i.e., be a G protein- or βarrestin-“biased” receptor, depending on the bound ligand. A plethora of experimental evidence suggests that the GLP-1R does not undergo desensitization in physiologically relevant tissues in vivo, but rather, it produces robust and prolonged cAMP signals. A particular property of constant cycling between the cell membrane and caveolae/lipid rafts of the GLP-1R may underlie its lack of desensitization. In contrast, GIPR signaling is extensively mediated by βarrestins and the GIPR undergoes significant desensitization, internalization, and downregulation, which may explain why both agonists and antagonists of the GIPR exert the same physiological effects. Here, we discuss this evidence and make a case for the GLP-1R being a phenotypically or functionally G_s_-selective receptor. We also discuss the implications of this for the development of GLP-1R poly-ligands, which are increasingly pursued for the treatment of obesity and other diseases.

## 1. Introduction

Glucagon-like peptide-1 (GLP-1) receptor agonists (GLP1-RAs) have recently emerged as one of the most effective classes of pharmaceuticals for beneficial modulation of metabolic pathways, suppression of appetite/food intake, cardiovascular risk reduction, and action against obesity (weight loss induction) [1]. They also appear to have a plethora of beneficial effects in many different organs/systems in the body, in large part because they reduce adiposity and its closely associated chronic low-grade inflammation, which can precipitate or complicate almost every chronic disease known to man [2]. GLP1-RAs work against obesity via a combination of various mechanisms and physiological effects, including suppression of food/energy intake by causing food/taste aversion in the central nervous system (CNS) [3]. Another important effect of these drugs conducive to weight loss is their lipolytic action in white adipocytes (white fat cells), the body’s fat storage cells. These white adipocytes play a crucial role in energy homeostasis and are central to the pathophysiology of obesity, since their hypertrophy (adipocyte enlargement) or hyperplasia/proliferation (formation of new fat cells) leads to an increase in adipose tissue mass or fat volume expansion, directly increasing body weight [4]. The profound therapeutic impact of the pleiotropic actions of these medications was very recently demonstrated in a study that found additional benefits for tens of different health outcomes conferred in diabetics by the addition of these drugs to the usual (standard) care, which included even the most up-to-date guideline-recommended medication classes, such as the sodium/glucose co-transporter (SGLT)-2 inhibitors [5].

Being a peptide, GLP-1 cannot enter the cell, and thus utilizes a single G protein-coupled receptor (GPCR) residing at the plasma membrane to exert its effects in cells, the GLP-1 receptor (GLP-1R) [6]. GLP-1R is a class B1 (secretin/glucagon subclass) GPCR that couples to the G_s_ (stimulatory) type of heterotrimeric G proteins, which activates adenylyl cyclase (AC) to synthesize the second messenger cyclic or 3’-5’-adenosine monophosphate (cAMP) [7]. Based on the observation that several other class B1 GPCRs exert many of the same physiological effects in vivo, particularly in terms of glucose-dependent insulin secretion (incretin action) and reduction in adiposity/obesity, several agents have been developed (and continue to develop) to activate other glucagon family receptors in addition to the GLP-1R for enhanced therapeutic effect [1]. The glucose-dependent insulinotropic peptide receptor (GIPR), the glucagon receptor (GCGR), and the amylin receptors (AMYRs) are the most important examples of these additional GPCRs targeted for therapeutic benefit. This has led to the advent of dual agonists, i.e., agents that co-activate the GLP-1R and one more receptor at the same time (e.g., GLP-1R/GIPR co-agonists, GLP-1R/AMYR co-agonists), and of triple unimolecular agonists, activating two more receptors in addition to the GLP-1R simultaneously (e.g., GLP-1R/GIPR/GCGR co-agonists) [1]. However, despite the seemingly straightforward rationale behind boosting the potency and/or efficacy of a therapeutic agent by targeting, with a single molecular entity, as many receptors that exert the same effect via same or similar signaling pathways as possible, the reality of the phenotypic results is more complicated.

This perspective examines important features of GLP-1R signaling, uncovered in several studies over the past decade or two, which may distinguish it from that of some other closely related receptors within the glucagon family, such as the GIPR. In addition, it highlights qualitative (and quantitative) differences in the level of functional selectivity or “bias” of the signaling of GLP-1R vs. that of GIPR and discusses the accumulating evidence in the literature suggesting that the GLP-1R displays natural functional selectivity for G_s_ protein signaling, with the caveat, of course, that signaling bias always depends on the ligand activating the receptor. Finally, the implications of these signaling selectivity differences for the pharmacology and therapeutic use of glucagon family multi-receptor agonists (poly-ligands) and GLP-1R mono-agonists are discussed.

## 2. GLP-1R and Lack of Functional Desensitization In Vivo

GLP-1R, similar to several other glucagon family GPCRs, including the prototypic for this family glucagon receptor itself, couples to the G_as_ subunit of heterotrimeric G proteins [7]. In fact, GLP-1R seems to couple exclusively to this signal transducer, as it consistently results in cAMP increases inside cells in every study published to date on GLP-1R signaling. There is a very limited amount of literature suggesting that GLP1-RAs can induce intracellular Ca^2+^ signaling and that GLP-1R can couple to G_q/11_ proteins, which activate phospholipase C (PLC)-β with subsequent elevation in intracellular free calcium concentration [8,9]. However, these studies were carried out in transfected cells heterologously overexpressing the receptor at supraphysiological levels. Additionally, GLP-1R can activate PLC and Ca^2+^ signaling without coupling to G_q/11_ proteins, simply through the actions of the cAMP effectors protein kinase A (PKA) and exchange protein directly activated by cAMP (Epac)-1/2, both of which affect a vast number of Ca^2+^-handling proteins and channels in every cell [10]. Thus, the Ca^2+^ signaling properties that GLP-1R agonism imparts could still be cAMP-dependent. Finally, engagement of a receptor with a G protein does not necessarily translate into activation of the latter; any given GPCR can interact with any G_a_ subunit in its vicinity at any given time, but the interaction might be insufficient to result in nucleotide exchange, i.e., in G_a_ activation. Indeed, the calcitonin receptor, which is also a class B1 G_s_-coupled GPCR, similar to the GLP-1R, has been shown to possess different GEF (guanine nucleotide exchange factor) activities when bound to slightly different agonists (human vs. salmon calcitonin) due to different G_as_ residency times on the receptor, translating into vastly different agonist efficacies [11]. This highlights the importance of the receptor’s agonist-induced GEF activity, i.e., actual G protein activation upon coupling, over its mere agonist-induced interaction with a G protein.

Not only does GLP-1R appear to signal exclusively through the G_s_ protein/AC/cAMP axis, but it also produces a rapid, robust, and prolonged cAMP signal that does not drop in intensity for at least 5–6 h (!) post-agonist application [7,12]. This property of sustained in time cAMP production has actually been known for glucagon since the early 1970s (GLP-1 was not known at the time) and in the studies by Sutherland et al., as well as Rodbell and Birnbaumer et al., on liver AC and its activation by glucagon and other hormones, i.e., years before the discovery of G proteins and GPCRs [13,14]. In perfused rat livers, glucagon started increasing cAMP levels within 30 s, induced a 60-fold (!) peak increase over basal within 3 min, and cAMP levels declined very slowly thereafter [13]. As shown by Kobilka et al. [7], GLP-1R behaves in the exact same manner as the glucagon receptor in terms of G_s_ protein activation. That same study also examined the mechanisms for the robust and sustained G protein activation by class B GPCRs, as opposed to the less robust and much shorter-lived G protein activation by class A GPCRs [7]. Studying the activation kinetics of the GCGR, as well as the GLP-1R and other class B G_s_-coupled receptors, the authors reported a very important distinction for class B vs. class A GPCRs in the relationship between the transmembrane (TM) helix TM6 outward movement, the hallmark step of GPCR activation, and G_a_ subunit interaction [7]. This distinction is responsible for differences in the free energy barriers separating the various activation states (inactive, intermediate, and fully active) of class B vs. class A GPCRs [7]. Specifically, G_as_ subunit interaction, induced by agonist binding, precedes the TM6 opening of GCGR and (probably of) GLP-1R, whereas the inverse is true for class A GPCRs, i.e., TM6 opening is a prerequisite for G_a_ subunit interaction and activation [7]. This striking difference is probably a result of the fact that agonists bind class B GPCRs predominantly at their large extracellular N-termini and TM1 helices, whereas class A GPCRs mainly interact with agonists deep inside the TM helical bundle and/or at the extracellular loops (ECLs) [15]. This essentially means that, whereas agonist binding induces TM6 opening in a class A GPCR with simultaneous G_a_ interaction and almost instant nucleotide release, i.e., G_a_ subunit activation, agonist binding alone at a class B GPCR induces G_a_ subunit interaction without inducing, necessarily, TM6 opening, which is a prerequisite for the guanine nucleotide exchange on the G_a_ subunit, i.e., G protein activation, to occur [7]. Thus, the GEF activity of GLP-1R (and of GCGR) is slower than that of class A GPCRs, and agonist binding alone does not decrease the free energy of activation of GLP-1R by much [7]. On the other hand, TM6 of class B GPCRs relaxes back to the “closed” state very slowly, which means that, once the TM6 of GLP-1R has “opened” and G_as_ activation has started, it remains open for a long time, allowing for rapid activation of G_as_ upon the receptor’s subsequent encounters with G proteins [7]. This mechanism of activation has two very important biochemical, pharmacological, and physiological consequences for the GLP-1R: a) The agonist-bound intermediate and fully active states of the receptor are separated from the inactive (agonist unbound, “apo”) state by a large energy barrier, meaning that GLP-1R has zero constitutive activity and the duration of its signaling through cAMP is completely dictated by the amount of time it is bound to the agonist; and b) Once activated (with its TM6 “open”), GLP-1R can activate the G_as_/AC/cAMP signaling axis robustly and continuously, hence the robust and prolonged cAMP signals GLP-1R (and GCGR) can produce; in other words,, the GEF activity of GLP-1R for G_as_ subunits, albeit slow initially, is sustained and cumulatively higher over time [7,12,13,14] (Figure 1).

However, this mechanism for GLP-1R-induced G_s_ protein activation alone is not sufficient to explain its remarkably sustained cAMP signaling in cells, since the vast majority of GPCRs undergo agonist-dependent desensitization, i.e., G protein decoupling [16]. Receptor phosphorylation by GPCR-kinases (GRKs) and subsequent βarrestin binding, which usually (albeit not always) leads the receptor to internalize in endosomes, are responsible for this process [17,18,19,20]. The sustained cAMP signaling elicited by the GLP-1R clearly indicates that this receptor is somehow resistant to GRK/βarrestin-dependent desensitization (Figure 1).

Indeed, several lines of evidence suggest that GLP-1R signaling and function in vivo are unaffected by βarrestins, suggesting that βarrestins affect neither GLP-1R desensitization (G protein coupling efficiency), nor GLP-1R internalization (Figure 2). More specifically, although it can undergo short-term homologous (GLP-1-induced) desensitization in vitro [21], the GLP-1R effects on glucose homeostasis appear unaffected by repeated agonist administrations in vivo [21] and its effects in patients and in animal models in vivo do not diminish to any significant degree over time [22,23,24,25,26,27]. Even acutely in vitro, GLP-1R desensitization is evident only after 1–2 h of pretreatment and only upon re-challenge with the considerably high concentration of 100 nM GLP-1 [21]. Notably, GLP-1R appears resistant to heterologous desensitization, as well [21], although this probably holds true for its desensitization by other G_s_-coupled receptors only, as phorbol esters and protein kinase C (PKC) have been reported to phosphorylate and desensitize the GLP-1R [21,28], which means that certain G_q_-coupled receptors can desensitize it.

Additionally, a study in cultured pancreatic beta cells reported that βarrestin1, although it can interact with the GLP-1R, does not affect its desensitization nor internalization, and, in fact, βarrestin1 knockdown actually reduced cAMP production by the GLP-1R [29]. This study is notable for an additional reason: it is the only study to date that actually shows direct interaction of the GLP-1R with a βarrestin via co-immunoprecipitation, rather than simple βarrestin translocation to the membrane or interaction inferred by fluorescence-tagged protein proximity assays (FRET or BRET). However, the GLP-1R-βarrestin1 interaction shown in that study could be agonist-independent, as it was detected even in the absence of GLP-1 [29], and thus not conducive to GLP-1R homologous desensitization, which would be consistent with the authors’ findings on cAMP production. Moreover, extracellular signal-regulated kinase (ERK) activation by GLP-1R is predominantly PKA (i.e., cAMP)-dependent in transfected HEK293 cells [30] and, importantly, both the blood glucose-lowering effect of GLP-1 in βarrestin2-knockout mice and the cAMP-producing potency and efficacy of GLP-1 in βarrestin1/2-knockout cells are intact [31]. Finally, the insulinotropic actions of GLP-1 are preserved even in diabetic patients, although type 2 diabetes diminishes the same actions that GIP exerts in humans [30,32]. Taken together, these studies demonstrate the relative inability of GLP-1R to undergo desensitization, i.e., to uncouple from G_s_ proteins and cAMP production, neither in vivo (in animal models and humans), nor in cultured cells expressing physiological levels of native GLP-1R in vitro (Figure 1 and Figure 2).

A possible mechanism for the apparent lack of significant desensitization of the GLP-1R, despite its ability for phosphorylation by GRKs and to interact with βarrestins [33,34], could be the receptor’s reportedly continuous “cycling” between the cell membrane and early endosomes in the cytoplasm via dynamin- and caveolin-1-dependent processes [35,36]. This caveolae-mediated “shuttling” of the GLP-1R, which is indispensable for proper signaling and function of the receptor, is GRK- and βarrestin-independent, since GLP-1R contains a caveolin interaction motif in its second intracellular loop and thus, directly interacts with caveolin [35]. This trafficking pattern may “shield” GLP-1R from GRK/βarrestin-mediated desensitization at the cell membrane, and, perhaps more importantly, can readily replenish those GLP-1R molecules that are desensitized, i.e., GRK (or PKC)-phosphorylated and uncoupled from G_s_ proteins, with new receptor molecules coming via caveolae from early endosomes back to the plasma membrane ready to activate more G_as_ subunits (Figure 1). In other words, GLP-1R is capable of very fast resensitization, which translates into a minimal overall desensitization phenotype over a period of a few hours post-agonist activation. Further corroborating the potential protection against GRK/βarrestin-mediated desensitization via continuous trafficking between the plasma membrane and the cytoplasmis another study showing that the interaction of βarrestin2 with the GLP-1R is very transient, i.e., the GLP-1R is a class A GPCR in terms of arrestin binding [37]. Perhaps the agonist-induced interaction with βarrestins at the membrane is too short-lived to confer significant GLP-1R desensitization, due to the rapid caveolin-mediated GLP-1R internalization upon agonist activation.

In contrast to its desensitization, whose degree of dependence on βarrestins is still a matter of debate, GLP-1R internalization has been consistently and unequivocally shown to be βarrestin-independent (Figure 2, left) but highly G protein-dependent [31,33,34,35,37,38,39], as well as dynamin-, clathrin-, caveolin- and, intriguingly, GRK-dependent [33,34,35]. A very interesting study reported that, as GLP-1R continuously recycles back-and-forth between the cell membrane and early endosomes via caveolae (see above), it remains bound to the agonist and signals through G_s_ proteins and cAMP the whole time [12] (Figure 1). Additionally, only a small percentage of the internalized GLP-1R is directed to the lysosomes for degradation and very slowly, with a half-life of over 2 h [12] (Figure 2, left). This, coupled with its resistance to desensitization, would effectively translate into sustained cAMP signaling by the GLP-1R, given that, as mentioned above, the time duration of agonist binding dictates the time length of GLP-1R activation. Notably, this signaling pattern could be akin to what was very recently reported for the vasopressin V_2_ receptor, which also displays βarrestin-independent, sustained G_as_ signaling when stimulated with arginine vasopressin [40]. Finally, a very recent proteomic study comparing agonist-induced phosphorylation patterns among GLP-1R, GCGR, and GIPR, demonstrated that GLP-1R undergoes minimal, barely detectable (~15%) agonist-induced phosphorylation leading to βarrestin recruitment, but not to desensitization or internalization [41], thereby further reinforcing the notion of the resistance of GLP-1R to homologous desensitization and the actions of βarrestins.

In conclusion, thanks, on the one hand, to its inherent mechanism of prolonged G_s_ protein activation, and, on the other hand, to its resistance to βarrestin-mediated desensitization and internalization, GLP-1R produces robust and prolonged cAMP signals inside cells (Figure 1). Furthermore, the fact that it only activates the G_s_/AC/cAMP signaling axis makes the GLP-1R a phenotypically (functionally) G_s_ protein-selective (“biased”) receptor in vivo, at least in native cells and tissues (Figure 2).

## 3. GLP-1R vs. GIPR: Similar Receptors, Distinct Signaling Properties

Many of the aforementioned studies that provide substantial evidence for the G_s_ protein “bias” of GLP-1R signaling studied the same signaling properties of the closely related GIPR in parallel and reported interesting and striking distinctions between these two glucagon family receptors. Although phylogenetically, structurally, and functionally closely related, GLP-1R and GIPR appear to display distinct regulation of their signaling. The most important difference is in their desensitization (agonist-dependent G_s_ protein decoupling) properties and reliance on βarrestins for signaling (Figure 2). In contrast to GLP-1R, GIPR internalization is significantly dependent on βarrestins [30,38,42,43,44,45]. Moreover, the blood glucose-lowering effect of the GIPR is diminished in βarrestin2-knockout mice and, quite interestingly, the cAMP-producing potency and efficacy of GIP in βarrestin1/2-knockout cells are also significantly diminished, rather than enhanced, indicating an essential role for βarrestins in normal GIPR signaling towards cAMP production [42,46] (Figure 2). GIPR has been demonstrated to undergo significant functional desensitization, internalization, and downregulation, with severely desensitized capacity to produce cAMP in transfected cells, as well as in primary adipocytes, neuronal cells, and pancreatic islet cells in vitro [43]. Its capacity to stimulate lipolysis in adipocytes and its functions in white adipose tissue and adrenal glands are also desensitized in diet-induced obesity mouse models in vivo [43]. Prolonged agonist exposure leads to GIPR functional desensitization in isolated pancreatic islets, as well [47]. Although inter-species differences in GIPR desensitization properties may exist [48], it has become abundantly clear that GIPR becomes, over time, desensitized also in humans in vivo, since the blood glucose-lowering effect of GIP is impaired in diabetic patients, despite endogenous GIP levels being normal or even elevated [32,49,50,51]. In stark contrast, GLP-1RAs display stable efficacy in patients over time [21,22,23,24,25,26,27,32]. In fact, GIPR desensitization appears so severe that antagonism of this receptor phenocopies its agonism [42,51]. Indeed, dual GLP-1 ligands that activate the GLP-1R but simultaneously block the GIPR (AMG133 or maridebart cafraglutide, MariTide^®^) are currently in clinical trials for obesity/weight loss and seem to work as well as (if not even better than) dual GLP-1R/GIPR agonists, such as tirzepatide (Zepbound^®^) (see below) [52].

Notably, GIPR depends on βarrestin interactions for signaling. Indeed, recent studies demonstrate the essential role specifically of βarrestin2 in the insulinotropic action of the GIPR [47] as well as the differential roles of βarrestin2 in GIPR vs. GLP-1R signaling [45,46]. Interestingly, even the signaling of GIPR toward cAMP synthesis seems to depend on βarrestins, but in the opposite manner, to the one theoretically expected: efficacy of GIPR at producing cAMP diminishes, instead of increasing, in the absence of βarrestins [42], which indicates a profound effect of βarrestins on GIPR signaling beyond receptor desensitization (Figure 2, right). The nature of this effect remains to be determined; however, it should be noted here that, in terms of cAMP production, both the potency and efficacy of the natural agonist (GIP)-activated GIPR are significantly lower than the potency and efficacy of natural agonist (GLP-1)-activated GLP-1R in cells [42,53] (Figure 2). Indeed, in transfected HEK293 cells expressing the same levels of GLP-1R or GIPR, and at two vastly different receptor densities (obtained with either 0.5 mg or 10 mg receptor cDNA transfections), it was determined that GLP-1R induced the formation of approximately double the amount of cAMP the GIPR induced, with the surface expression (B_max_) levels being equal for both receptors [42]. In other words, the same number of GLP-1Rs produced twice as much cAMP as the same number of GIPRs [42] (Figure 2). This is consistent with a somewhat older study on tirzepatide (see below), which also reported slower and vastly lower cAMP production for GIP than for GLP-1 at every (equal) concentration of these hormones tested [53]. Whether this reflects the impact of βarrestins on GIPR signaling or another fundamental difference in the mechanism of activation between these two receptors is an open question that awaits delineation in future studies. Notably, GIPR seems to negatively affect the signaling and function of GLP-1R in cells (e.g., neuronal sub-populations), in which both receptors co-exist [52,54]. Additionally, the agonist-independent (basal) activity of GIPR is much higher than that of GLP-1R, which essentially displays no constitutive activity [37]. Finally, it should also be pointed out that, although the mechanism of G_s_ protein activation by GCGR and GLP-1R discussed above should theoretically be shared by all class B GPCRs, GIPR was not among the GPCRs studied in that seminal work of Kobilka et al. [7]. In contrast, a recent study on the binding modes of the dual agonist tirzepatide to the two receptors illuminated important differences in the manner GLP-1R and GIPR are activated by tirzepatide and by their native agonists [55]. It is thus quite plausible that significant differences in the GEF activities for G_s_ exist between GIPR and GLP-1R, which could also explain the higher potency of the latter at stimulating cAMP production (see above) [42].

In conclusion, GLP-1R signals exclusively via G_s_ proteins (G_s_–selective) and does not undergo significant desensitization, in contrast to GIPR that interacts with both G_s_ proteins and βarrestins equally robustly (non-selective, pleiotropic balanced receptor) and is severely desensitized in vivo. In addition, GLP-1R stimulates AC and cAMP synthesis more efficiently than GIPR (Figure 2).

## 4. Implications for Glucagon Family Receptor Poly-Agonists: GLP-1R Agonism Is Sine Qua Non

Over the past few years, novel class B1 GPCR poly-ligands, i.e., compounds that bind more than one receptor simultaneously, have joined the ranks of the classic GLP-1RAs (GLP-1R mono-agonists), such as exenatide, liraglutide, semaglutide, dulaglutide, etc. Tirzepatide is the first dual GLP-1R/GIPR agonist to be FDA-approved for diabetes and obesity/overweight [1]. Several more dual receptor agonists are currently in the pipeline or in clinical trials and the list now includes triple receptor agonists, such as retatrutide (GLP-1R/GIPR/GCGR tri-agonist [56]), but also poly-ligands with interesting properties, such as maridebart cafraglutide (AMG133), which is a dual GLP-1R agonist/GIPR antagonist [52]. Notably, the only property all these poly-ligands share is that they all activate the GLP-1R, albeit with varying potencies (Figure 3). Thus, it seems that, no matter what a ligand does to other receptors (activation or blockade), GLP-1R activation is an indispensable property for therapeutic effect. This may reflect the powerful capability of GLP-1R to stimulate cAMP formation inside the cell.

The very efficient G_s_ protein coupling and cAMP production elicited by GLP-1R activation may also offset the impact of the decreased potency. In terms of cAMP production, dual and triple agonists display at the GLP-1R compared to “classic” GLP-1R mono-agonists (Figure 3). Although the relationship between the number of receptor types stimulating cAMP formation and the amount of cAMP formed in response to any given agonist might be expected to be linear, this does not hold true in the living cell, for two main reasons. One, as known for decades, the G_s_ protein/AC system is saturable: this discovery was, in fact, what led Rodbell, Birnbaumer, and their colleagues, almost 60 years ago, to abandon the belief that AC itself was the receptor for the hormones [57]. The other salient reason is that various receptors crosstalk with each other, thereby affecting each other’s cAMP producing capacity. Indeed, almost all poly-agonists studied to date seem to be less potent at GLP-1R-dependent cAMP formation compared to GLP-1 or even a “classic” GLP1-RA (GLP-1R mono-agonist) [53,55,56]. For instance, tirzepatide is a full agonist for the GIPR (equal potency with GIP) but a quite weak agonist, essentially a partial agonist, at the GLP-1R (much less potent than GLP-1) [53,55] (Figure 3). Retatrutide has a much higher potency at the human GIPR than GIP, in terms of both cAMP production and lipolysis in human adipocytes, i.e., acts as a “super” agonist for the GIPR, but is only a partial agonist for the GCGR and, again, for the GLP-1R [56] (Figure 3). Even maridebart cafraglutide, which binds but does not activate GIPR, is also a partial agonist for the GLP-1R [52], which may reflect the fundamental interplay between receptors competing for activation of the same enzyme (AC) in the same cell (Figure 3). Given that GIPR antagonists cause retention of the receptor at the plasma membrane (prevent GIPR internalization [43]), perhaps the mere act of “stabilizing” the GIPR at the cell membrane makes maridebart cafraglutide less potent at producing cAMP via the GLP-1R than it would be if it did not interact with the GIPR at all.

Taken together, these findings indicate the large variability in the cAMP responses these poly-ligands induce, which, in turn, may explain clinical differences among them, such as the very recently reported differences in their clinical efficacies at reducing body weight, with retatrutide appearing the most effective agent (producing a ~24% weight loss, almost equivalent to surgical interventions) [58]. On the other hand, all of them appear effective therapeutically despite their variable cAMP responses and partial agonist activity at the GLP-1R. This is probably due to a) the fact that the cAMP response is highly cell/tissue type-dependent; and b) activation of additional G_s_-coupled receptors may compensate for the lower potency at GLP-1R. However, the robustness of GLP-1R-G_s_ protein coupling efficiency may also play a role; since GLP-1R induces cAMP formation very efficiently, even a mildly potent GLP-1R agonist can elicit sufficient cAMP accumulation for a physiologically or clinically meaningful effect to occur. The only requirement appears to be the ability to activate the GLP-1R, i.e., being a GLP-1R agonist (even a weak/partial one).

## 5. Conclusions

The notions that GLP-1R is functionally G_s_ protein-selective, lacks constitutive activity, and cannot undergo G protein uncoupling while agonist-bound indicate that every cell that expresses this receptor has essentially only three “tools” to regulate its activity. One tool is the regulation of levels of GLP-1, the natural endogenous agonist hormone. For example, systemic inflammation, via tumor necrosis factor (TNF)-a and its type 1 receptor (TNFR1) inhibits GLP-1 secretion from L cells and reduces circulating GLP-1 levels in the blood [59], probably in an effort to promote inflammation, since GLP-1R has anti-inflammatory properties [6]. Another tool is the regulation of cAMP-degrading phosphodiesterase (PDE) activity, which is probably the only level of regulation downstream of GLP-1R. GRKs do not affect receptor desensitization and there are no known GAPs (GTPase-activating proteins) that inactivate G_s_ proteins to date, which means that there are no regulators downstream of GLP-1R and upstream of cAMP that can affect GLP-1R signaling. Therefore, regulation of cAMP itself, either via modulation of AC activity (cAMP synthesis) or PDE activity (cAMP degradation) appears to be the only possibility to fine-tune GLP-1R signaling at the post-receptor level. Indeed, the cAMP-specific PDE4 inhibitor rolipram was able to increase the sustained GLP-1R signaling to cAMP in vitro both in the absence and presence of βarrestins [45] and roflumilast, another PDE4 inhibitor, can enhance GLP-1R anti-inflammatory signaling in cardiac cells [60].

The third and perhaps the most important physiological tool is the regulation of GLP-1R gene expression, i.e., transcriptional upregulation or downregulation of this receptor. Indeed, cell surface expression (i.e., GLP-1R B_max_) appears to be the single most critical determinant of GLP-1R function in vivo [39] and GLP-1R upregulation or downregulation via modulation of gene expression seems to occur during physiological and pathophysiological conditions in various tissues and cell types. For example, GLP-1R expression appears to be upregulated in macrophages transitioning from the M1 (pro-inflammatory) to the M2 (anti-inflammatory) polarization phenotype [61]. GLP-1R expression changes, usually downregulation (but also upregulation in some cases), have also been reported in various anatomical parts of adipose tissue and in other non-fat tissues during large changes in fat volume and during body weight gain or loss in humans, e.g., in obesity, insulin resistance, after bariatric surgery, etc. [62,63,64,65].

The amount of research in the fields of incretin receptor biochemistry and pharmacology has skyrocketed in recent years, owing to the huge popularity and big cultural impact of Ozempic^®^ and related GLP1-RAs as (potentially) revolutionary medications for obesity and weight loss assistance. Despite this explosion in research, there is so much more that we still do not know about this medication class and the research interest in them has probably not even peaked yet. One of the most important questions regarding GLP-1R pharmacology still awaiting an answer is the precise role of βarrestins in GLP-1R signaling. There is a plethora of studies indicating the interaction of the receptor with βarrestin1 or βarrestin2; yet, GLP-1R does not undergo desensitization by βarrestins, it does not internalize via βarrestins, and its in vivo function is unaffected by βarrestins [42]. Further complicating the picture, a couple of studies suggest enhancement of GLP-1R cAMP signaling by βarrestins [29,45]. Perhaps this conundrum arises from the fact that almost all these studies on βarrestin interaction are bioluminescence-based and carried out in heterologous (transfected) cells overexpressing tagged (non-native) GLP-1R and βarrestins. As mentioned above, the only study we could find in the literature that shows the actual GLP-1R–βarrestin interaction with a biochemical method (co-immunoprecipitation) was the one published by Sonoda et al. in 2008, and even in that study the GLP-1R–βarrestin interaction might have been agonist-independent [29]. This raises the intriguing possibility that the numerous new GLP-1R agonist compounds being developed for enhanced efficacy against obesity, all of which invariably prove to be G_s_/cAMP-“biased” agonists upon pharmacological characterization (which is also perhaps why they have enhanced weight loss efficacy) [9,53,66,67], turn out to be G_s_/cAMP-“biased” exactly because native GLP-1R is inherently G_s_/cAMP-“biased”. Perhaps, differences in potency and/or efficacy compared to the natural agonist (GLP-1) are what makes them behave as G_s_/cAMP-“biased” versus GLP-1.

Given the astounding plethora of prospective indications for GLP-1R agonists, including diseases not necessarily or readily associated with metabolic abnormalities, from behavioral/addictive disorders (addiction, opioid use disorder, alcohol use disorder, etc.) to even glaucoma, arthritis, and cancer [68,69,70,71,72,73,74,75,76,77], the level of research intensity in this field is bound to keep steady or even further increase. GLP-1RAs will probably end up not fully living up to their current hype, but they have already proven that they are no “busts”. As is always the case, the truth is somewhere in the middle. In any case, nature seems to have endowed GLP-1R with a tremendous ability to stimulate cAMP production inside mammalian cells. This feature may confer unique physiological advantages to this receptor over other, even closely related, GPCRs, such as the GIPR. In light of this, the fact that GLP-1R antagonists appear devoid of any therapeutic value may seem not all that surprising. The pharmacological perspective this article presents, i.e., that the GLP-1R is a remarkably efficient G_s_ protein/AC-selective (“biased”) receptor, gleaned from carefully conducted pharmacological studies over the past 10–15 years, will hopefully assist in the design and development of better, safer, and even more effective medications for the numerous ailments whose treatments GLP-1R can positively impact.

## Figures and Tables

**Figure 1 ijms-26-07239-f001:**
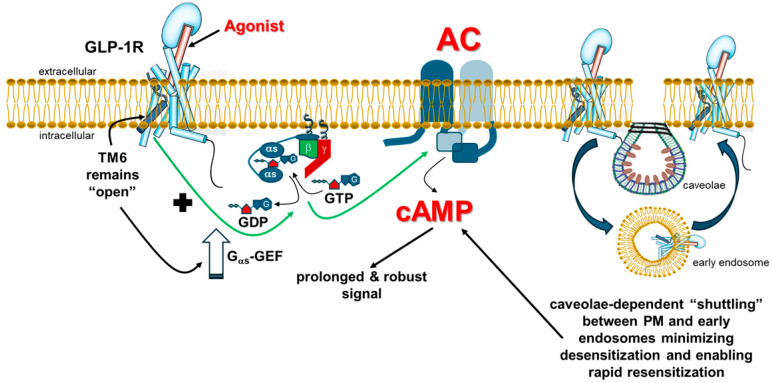
Mechanisms for the robust and prolonged cAMP signaling by the GLP-1R. Left: Because, once activated, the TM6 of agonist-bound GLP-1R remains open for a long time, the receptor can stimulate the GDP-to-GTP exchange on the G_as_ subunit in a sustained manner, i.e., GLP-1R displays enhanced (over time) G_as_-GEF activity (modified from Ref. [7]). Right: Agonist-bound GLP-1R appears to “shuttle” back-and-forth between the PM and early endosomes for a prolonged period of time post-agonist activation in a caveolin-dependent manner, which may “shield” the receptor from extensive GRK/βarrestin-mediated desensitization at the cell membrane. This cycling between the membrane and the cytoplasm may also provide a mechanism for prompt resensitization of the GLP-1R, via replacement of desensitized receptors at the membrane by new ones returning from the cytoplasm ready to activate more G_as_ molecules. Combined, these two mechanisms (high receptor GEF activity and continuous cycling between membrane and cytoplasm) could explain the robust and prolonged cAMP signals produced by GLP-1R. Abbreviations: as: G_as_ subunit; βγ: The Gβγ subunit dimer of heterotrimeric G proteins; PM: Plasma (cell) membrane.

**Figure 2 ijms-26-07239-f002:**
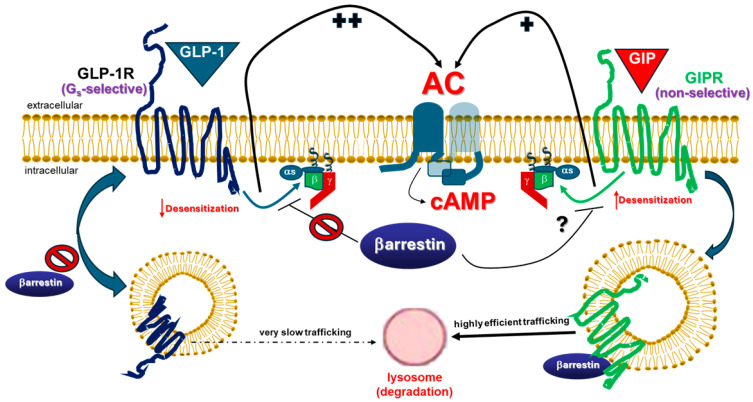
Distinct signaling properties of the G_s_-selective GLP-1R vs. the non-selective GIPR. GLP-1R is more potent than GIPR at stimulating AC and producing cAMP. Although it can interact with βarrestins to induce signaling, GLP-1R undergoes minimal (if any) decoupling from G_as_ (desensitization) and traffics to lysosomes for degradation and downregulation very slowly, independently of βarrestins. These processes result in robust and prolonged Gs protein activation and cAMP signaling from the GLP-1R. In contrast, GIPR depends on βarrestins for signaling towards effectors, including, potentially, even G_as_/AC stimulation. Additionally, the GIPR–βarrestin complex rapidly internalizes into endosomes (without recycling to the cell membrane) and GIPR traffics to lysosomes for degradation and downregulation with high efficiency. These GIPR features result in weaker and short-lived cAMP signaling from this receptor. The curved double arrow symbol indicates bidirectional recycling; 
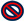
 indicates no effect of βarrestin; “+” symbol next to the black arrow indicates potentiation; “?”: Exact effect of βarrestin on Gs coupling not known; “++” symbol indicates higher potency than “+” symbol. Abbreviations: αs: G_as_ subunit; βγ: The Gβγ subunit dimer of heterotrimeric G proteins.

**Figure 3 ijms-26-07239-f003:**
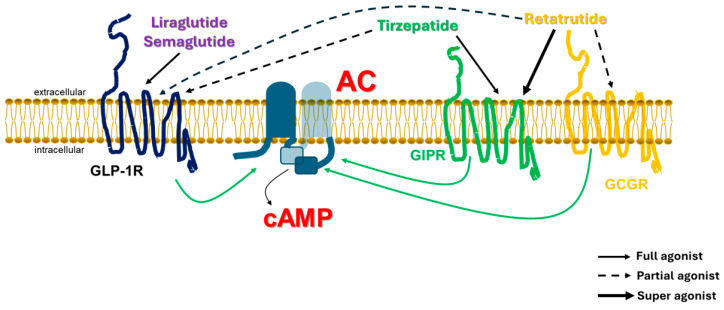
Differential potencies at GLP-1R agonism by various glucagon family receptor poly-agonists. Out of the various poly-agonists currently on the market or in development, only the GLP-1R mono-agonists (e.g., liraglutide, semaglutide) are full agonists at the GLP-1R with respect to cAMP synthesis.

## Data Availability

The raw data supporting the conclusions of this article will be made available by the authors on request.

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
