# Peer review of "Glucagon-like Peptide-1 Receptor (GLP-1R) Signaling: Making the Case for a Functionally Gs Protein-Selective GPCR"

_ijms, 2025, doi:10.3390/ijms26157239_

Round 1

Reviewer 1 Report

Comments and Suggestions for Authors

In their manuscript “Glucagon-like-peptide-1 receptor (GLP-1R) Signaling: Making the Case for a functionally Gs protein-biased GPCR” the authors review current studies on GLP-1R pharmacology. Especially highlighting the specificietes of GLP1-R compared to other members of the glucagon receptor family is of interest given the fact that dual agonists and triagonists are becoming more important.

However, in this section the review could be more detailed, since the authors compare GLP-1R to GIPR and highlight the distinct properties of these receptors. It would be beneficial if the authors would extent this part and also compare GLP-1R to GCGR and GLP-2R.

Furthermore, the term “biased” signalling refers to ligands inducing a specific signalling in GPCRs if the receptor is capaple of different signalling modes. The authors rather use this term to claim that GLP-1R almost exclusively signals via Gs-protein, thus, I found this wording misleading. I would suggest rephrasing to avoid misunderstanding. Furthermore, calling GIPR unbiased might be confusing for readers familiar with biased agonism.

Figure 2: the arrows from GLP-1 / GIP to AC are misleading since not GLP-1 or GIP stimulate adenylyl cyclase.

Author Response

Comment #1: In their manuscript “Glucagon-like-peptide-1 receptor (GLP-1R) Signaling: Making the Case for a functionally Gs protein-biased GPCR” the authors review current studies on GLP-1R pharmacology. Especially highlighting the specificietes of GLP1-R compared to other members of the glucagon receptor family is of interest given the fact that dual agonists and triagonists are becoming more important.

However, in this section the review could be more detailed, since the authors compare GLP-1R to GIPR and highlight the distinct properties of these receptors. It would be beneficial if the authors would extent this part and also compare GLP-1R to GCGR and GLP-2R.

Author response: We thank this reviewer for the overall kind and positive comments about the quality of our work. This is an interesting suggestion made by the reviewer but, unfortunately, there are not a lot of studies (to our knowledge at least) on biased signaling by GCGR and GLP-2R, and definitely far less is known about biased signaling by those receptors than what is known about GLP-1R and GIPR biased signaling. This is why our perspective article has focused on the latter two receptors (GLP-1R and GIPR). We hope the reviewer understands.

Comment #2: Furthermore, the term “biased” signalling refers to ligands inducing a specific signalling in GPCRs if the receptor is capaple of different signalling modes. The authors rather use this term to claim that GLP-1R almost exclusively signals via Gs-protein, thus, I found this wording misleading. I would suggest rephrasing to avoid misunderstanding. Furthermore, calling GIPR unbiased might be confusing for readers familiar with biased agonism.

Author response: The reviewer raises an interesting point here. It is true that, traditionally, the term “biased signaling” refers to ligands, not receptors per se; however, this is exactly the point we try to make in our present article: that the GLP-1R behaves as a Gs/cAMP-“biased” receptor at all times, regardless of the activating ligand. Besides, the term “biased receptor” has started to gain some traction in the literature over the past few years, e.g., see Terry Kenakin`s review from 2019: “Biased Receptor Signaling in Drug Discovery”. Pharmacol Rev. 2019;71:267-315. As for GIPR, it is “unbiased” when activated by its natural agonist GIP. We hope this is clear to this reviewer now and apologize for any confusion.  

Comment #3: Figure 2: the arrows from GLP-1 / GIP to AC are misleading since not GLP-1 or GIP stimulate adenylyl cyclase.

Author response: Done, this is corrected now. Thanks for pointing this out to us.

Reviewer 2 Report

Comments and Suggestions for Authors

Remarks about the manuscript: 

The authors have presented a review article, "Glucagon-like peptide-1 receptor Signaling: Making the Case For a Functionally Gs Protein-biased GPCR." The article focuses on the biased signaling of the GLP-IR receptor compared to the GPIR receptors, highlighting their differences throughout the manuscript. This manuscript presents a carefully selected set of reference articles from the last two decades. This review provides a cautious explanation of the mechanistic and activation differences between the two receptors, with dependencies mediated by β-arrestins, and is supported by clear explanations. Relevant references comparing well-known developed agonists have been included, along with those from their early or clinical stage development.

Overall, the manuscript is well-set, with references to up-to-date information for GLP-IR receptors and their activation bias. However, I have some suggestions for improving the English sentences throughout the manuscript. There are too many “i.e.” uses in one sentence, and some sentences end with “in” or “is”. Example below

  1. GLP-1R is more potent at stimulating AC and producing cAMP than GIPR is. (figure2)
  2. Notably, GIPR seems to negatively affect signaling and function of GLP-1R in cells (e.g. neuronal sub-populations) both receptors co-exist in. (page 12, second sentence)

The present manuscript includes most of the data needed for publication in the IJMS journal; therefore, I recommend it for publication as is, after a check of the English sentences. 

This manuscript (ID: IJMS-3735758) can be accepted without revision.

Comments on the Quality of English Language

I have added a few examples in my review comments. 

Author Response

Comment #1: The authors have presented a review article, "Glucagon-like peptide-1 receptor Signaling: Making the Case For a Functionally Gs Protein-biased GPCR." The article focuses on the biased signaling of the GLP-IR receptor compared to the GPIR receptors, highlighting their differences throughout the manuscript. This manuscript presents a carefully selected set of reference articles from the last two decades. This review provides a cautious explanation of the mechanistic and activation differences between the two receptors, with dependencies mediated by β-arrestins, and is supported by clear explanations. Relevant references comparing well-known developed agonists have been included, along with those from their early or clinical stage development.

Overall, the manuscript is well-set, with references to up-to-date information for GLP-IR receptors and their activation bias. However, I have some suggestions for improving the English sentences throughout the manuscript. There are too many “i.e.” uses in one sentence, and some sentences end with “in” or “is”. Example below

  1. GLP-1R is more potent at stimulating AC and producing cAMP than GIPR is. (figure2)
  2. Notably, GIPR seems to negatively affect signaling and function of GLP-1R in cells (e.g. neuronal sub-populations) both receptors co-exist in. (page 12, second sentence)

Author response: We thank this reviewer for the overall kind and positive comments about the quality of our work. We have rephrased these sentences and proofread our text to correct all typos and improve the English of our manuscript.

Comment #2: The present manuscript includes most of the data needed for publication in the IJMS journal; therefore, I recommend it for publication as is, after a check of the English sentences. 

This manuscript (ID: IJMS-3735758) can be accepted without revision.

Author response: Again, we thank this reviewer for the positive comments about the quality of our work and are very pleased that he/she liked it.

Round 2

Reviewer 1 Report

Comments and Suggestions for Authors

I thank the authors for their reply to the comments, however, my main points still remain (refering to the initial comments):

Comment #1: The question remaining open at this point is, if the properties of GLP-1R are rather unique or is this something shared by the GCGR receptor family. Here, the authors demonstrate that GIPR is behaving differently, but to strengthen there conclusion more information on other incretin receptors would be needed.  

Comment #2: In the mentioned review, Kenakin does not use the term “biased receptor” but “biased receptor signalling”. Indeed, directly in the beginning of the Introduction Kenakin defines bias:

“When this term is applied to cellular signaling mediated by seven-transmembrane receptors (7TMRs), it refers to a pleiotropically linked receptor (one that is coupled to more than one signaling pathway) producing more of some of the signals at the expense of others.”  

If GLP-1R is regaredless of the activating ligand only interacts with and activates Gs protein it would be – applying Kenakin’s definition – monotonic / unbiased receptor signalling. I do understand the point the authors want to make, however, I think that the wording they use do not fit the definitions in the field.

Comment #3: Figure 2: AC is also not stimulated by the receptor but the G protein. With the arrows drawn outside of the cell this is not a correct presentation. Esp since the authors want to depict the fact that GLP-1R activation leads to a prolonged G protein activation.

Author Response

Comment #1: The question remaining open at this point is, if the properties of GLP-1R are rather unique or is this something shared by the GCGR receptor family. Here, the authors demonstrate that GIPR is behaving differently, but to strengthen there conclusion more information on other incretin receptors would be needed.  

Author response: We do not disagree with this point per se, but a) We feel discussion of other incretin receptors is beyond the scope of our perspective article, which is focused solely on showcasing the Gs bias of GLP-1R in stark contrast to the unbiased GIPR which lacks such bias; and b) As already mentioned in the previous round of review, we are not aware of studies on biased signaling of other glucagon family receptors. If the reviewer has any particular studies in mind we may have missed, relevant to our manuscript`s theme, we welcome his/her suggestions and we will do our best to cite and include them in a newly revised version of our discussion.

Comment #2: In the mentioned review, Kenakin does not use the term “biased receptor” but “biased receptor signalling”. Indeed, directly in the beginning of the Introduction Kenakin defines bias:

“When this term is applied to cellular signaling mediated by seven-transmembrane receptors (7TMRs), it refers to a pleiotropically linked receptor (one that is coupled to more than one signaling pathway) producing more of some of the signals at the expense of others.”  

If GLP-1R is regaredless of the activating ligand only interacts with and activates Gs protein it would be – applying Kenakin’s definition – monotonic / unbiased receptor signalling. I do understand the point the authors want to make, however, I think that the wording they use do not fit the definitions in the field.

Author response: We apologize but we do not understand the reviewer`s point here. If a receptor only activates one signaling pathway (in this case, GLP-1R only activates Gs to produce cAMP), how can it be unbiased? In our view at least, that`s the definition of a biased receptor. Also, the term "biased signaling" inherently refers to the receptor, since the receptor is the molecular entity that does the signaling, not the ligand. Finally, there are several examples of receptors referred to as "biased" receptors in the literature, e.g., see: "Pandey et al. Intrinsic bias at non-canonical, β-arrestin-coupled seven transmembrane receptors. Mol Cell. 2021;81:4605-4621.e11" or "Ishizuka et al. CXCR7 ameliorates myocardial infarction as a β-arrestin-biased receptor. Sci Rep. 2021 Feb 9;11(1):3426", etc. Therefore, the term "biased receptor" is anything but controversial or misleading.

Comment #3: Figure 2: AC is also not stimulated by the receptor but the G protein. With the arrows drawn outside of the cell this is not a correct presentation. Esp since the authors want to depict the fact that GLP-1R activation leads to a prolonged G protein activation.

Author response: Although the origins of the arrows were not meant to be taken literally, as this is just a cartoon, we understand the reviewer`s point and we have modified the figure accordingly. AC is activated by a receptor-stimulated Gs protein, so the arrows depict exactly that now in the revised Figure 2. We hope this now satisfies this reviewer.